# Predictors of mortality within the first year of initiating antiretroviral therapy in urban and rural Kenya: A prospective cohort study

Rachel A. Silverman[1,2,3]*, Grace C. John-Stewart[1,2,4,5], Ingrid A. Beck[6], Ross Milne[6], Catherine Kiptinness[2], Christine J. McGrath[2], Barbra A. Richardson[2,7], Bhavna Chohan[2,8], Samah R. Sakr[9], Lisa M. Frenkel[2,4,5,6,10], Michael H. Chung[1,2,4]

**1** Department of Epidemiology, University of Washington, Seattle, Washington, United States of America, **2** Department of Global Health, University of Washington, Seattle, Washington, United States of America, **3** Department of Population Health Sciences, Virginia Polytechnic Institute and State University, Blacksburg, Virginia, United States of America, **4** Department of Medicine, University of Washington, Seattle, Washington, United States of America, **5** Pediatrics, University of Washington, Seattle, Washington, United States of America, **6** Seattle Children's Research Institute, Seattle, Washington, United States of America, **7** Department of Biostatistics, University of Washington, Seattle, Washington, United States of America, **8** Kenya Medical Research Institute (KEMRI), Nairobi, Kenya, **9** Coptic Hospital, Nairobi, Kenya, **10** Department of Laboratory Medicine, University of Washington, Seattle, Washington, United States of America

* rsilverman@vt.edu

**Data Availability Statement:** Restrictions on data accessibility were imposed by the Kenyatta National Hospital Ethical Review Committee (KNH

## Abstract

### Introduction

Despite increased treatment availability, HIV-infected individuals continue to start antiretroviral therapy (ART) late in disease progression, increasing early mortality risk.

### Materials and methods

Nested prospective cohort study within a randomized clinical trial of adult patients initiating ART at clinics in urban Nairobi and rural Maseno, Kenya, between 2013–2014. We estimated mortality incidence rates following ART initiation and used Cox proportional hazards regression to identify predictors of mortality within 12 months of ART initiation. Analyses were stratified by clinic site to examine differences in mortality correlates and risk by location.

### Results

Among 811 participants initiated on ART, the mortality incidence rate within a year of initiating ART was 7.44 per 100 person-years (95% CI 5.71, 9.69). Among 207 Maseno and 612 Nairobi participants initiated on ART, the mortality incidence rates (per 100 person-years) were 12.78 (95% CI 8.49, 19.23) and 5.72 (95% CI 4.05, 8.09). Maseno had a 2.20-fold greater risk of mortality than Nairobi (95% CI 1.29, 3.76; $P = 0.004$). This association remained [adjusted hazard ratio (HR) = 2.09 (95% CI 1.17, 3.74); $P = 0.013$] when adjusting for age, gender, education, pre-treatment drug resistance (PDR), and CD4 count, but not when adjusting for BMI. In unadjusted analyses, other predictors ($P<0.05$) of mortality

ERC) to protect participant confidentiality and privacy due to the sensitive nature of the health information included in our analyses. Data collected from the study, including individual deidentified participant data and a data dictionary defining each field in the set, study protocol, statistical analysis plan, and informed consent form will be made available to others upon request after publication of this manuscript reporting study results and with a signed data access agreement. Data requests can be made to Engi Attia (eattia@uw.edu) at the Treatment Research and Expert Education (TREE) program in the Department of Global Health at the University of Washington.

**Funding:** This work was supported by grants from the National Institutes of Health [R01-AI058723 and R01-AI100037] (LMF), including an American Recovery and Reinvestment Act supplement [R01-AI058723] (LMF). Support of the study was provided by the University of Washington Center for AIDS Research [P30AI027757]. The Coptic Hope Center for Infectious Diseases is supported by the President's Emergency Plan for AIDS Relief through a cooperative agreement [U62/CCU024512] from the Centers for Disease Control and Prevention. The funders had no role in study design, data collection and analysis, decision to publish, or preparation of the manuscript.

**Competing interests:** The authors have declared that no competing interests exist.

included male gender (HR = 1.74), age (HR = 1.04 for 1-year increase), fewer years of education (HR = 0.92 for 1-year increase), unemployment (HR = 1.89), low body mass index (BMI<18.5 m/kg$^2$; HR = 4.99), CD4 count <100 (HR = 11.67) and 100–199 (HR = 3.40) vs. 200–350 cells/µL, and pre-treatment drug resistance (PDR; HR = 2.49). The increased mortality risk associated with older age, males, and greater education remained when adjusted for location, age, education and PDR, but not when adjusted for BMI and CD4 count. PDR remained associated with increased mortality risk when adjusted for location, age, gender, education, and BMI, but not when adjusted for CD4 count. CD4 and BMI associations with increased mortality risk persisted in multivariable analyses. Despite similar baseline CD4 counts across locations, mortality risk associated with low CD4 count, low BMI, and PDR was greater in Maseno than Nairobi in stratified analyses.

## Conclusions

High short-term post-ART mortality was observed, partially due to low CD4 count and BMI at presentation, especially in the rural setting. Male gender, older age, and markers of lower socioeconomic status were also associated with greater mortality risk. Engaging patients earlier in HIV infection remains critical. PDR may influence short-term mortality and further studies to optimize management will be important in settings with increasing PDR.

## Introduction

Substantial efforts have been made to accelerate diagnosis of HIV infection and start infected individuals on ART as soon as possible [1–3]. However, many HIV-infected individuals continue to delay testing and/or treatment until they are symptomatic with advanced HIV disease progression [4–6], increasing their risk of early mortality [7, 8]. Prior studies in sub-Saharan Africa and other settings have identified sociodemographic predictors of early mortality including male gender [7, 9–11] and older age [7, 9, 10]. Measures of lower socioeconomic status [12, 13] and single marital status [13] have also been identified in some, though not all [14, 15] studies that investigated these factors. Clinical predictors of mortality include low CD4 count [7, 9, 10, 16] and low body mass index (BMI), weight loss, and malnutrition [7, 16]. Pre-treatment drug resistance (PDR) was observed to impact longer-term mortality (>6 months post-ART initiation) in one study [17], but was not associated with mortality within 1 and 2 years post-ART in another [18]. In Kenya, some rural areas have lower rates of HIV testing, greater delays in treatment, higher HIV prevalence, higher HIV-related mortality [19–21], and greater burdens of other infections including diarrheal diseases, tuberculosis and other respiratory diseases, and malaria both generally and among HIV-infected individuals [19, 22–25].

The objective of this nested prospective cohort study was to assess the risk and predictors of short-term mortality among individuals participating in a randomized clinical trial (RCT) who initiated ART in 2013–2014 at two treatment clinics implemented by the same program (with the same clinical procedures and protocols), one in urban Nairobi, the capital city, and one in rural Maseno, Kisumu in Western Kenya. We examined sociodemographic and clinical correlates of mortality overall and across these sites. We hypothesized that older age, male gender, indicators of lower socioeconomic status, being enrolled at the rural vs. urban clinic, low CD4 count, low BMI, and PDR prior to ART initiation at study enrollment would correlate with increased mortality risk in unadjusted analyses. We also utilized multivariable regression

to assess the independent effects of these factors through exploratory analyses. By stratifying by treatment clinic site in all analyses, we examined potential differences in mortality correlates across the rural and urban locations. We ultimately aimed to gain greater understanding of factors driving short-term mortality risk among HIV-infected individuals initiating ART in high disease-burden areas in Kenya and similar settings.

## Materials and methods

### Study design, setting, and participants

This study was approved by the Human Subjects' Committees at Seattle Children's Hospital in Seattle, Washington (Institutional Review Board (IRB) Study #: 14124), and Kenyatta National Hospital in Nairobi, Kenya (Ethical Review Committee (ERC) Project #: P447/06/2016; approval reference #: KNH-ERC/A/297). All participants provided written informed consent prior to study enrollment as approved by the Human Subjects' Committees at Seattle Children's Hospital in Seattle, Washington, and Kenyatta National Hospital in Nairobi, Kenya.

We nested a prospective cohort study within a randomized clinical trial (RCT) investigating resistance testing-informed versus standard of care (SOC) treatment (RCT name: Oligonucleotide Ligation Assay (OLA) Resistance Study; ClinicalTrials.gov identifier: NCT01898754). Enrolled patients received care through the Coptic Hospital Hope Center for Infectious Diseases at three locations in Kenya, which provides HIV care [26, 27], standardized across clinic locations. For this RCT [28, 29], HIV-infected patients were enrolled from May 28th, 2013 to November 5th, 2014 at two clinics located in urban Nairobi (Ngong Road and Industrial Area) and one in rural Maseno, Kisumu. Participants received a CD4 test and health assessment through the Hope Center and were referred to the study if eligible for the RCT. Participants were followed for 12 months from ART initiation, either monthly or every two months per clinician discretion, and attended an exit visit at 15 months to receive their final OLA results. Eligibility criteria for the RCT included that participants were over two years of age, willing to initiate ART, and eligible to initiate ART based on Kenyan National Guidelines at the time of enrollment. The CD4 count threshold for ART eligibility from 2011 through mid-2014 was 350 cells/μL [30] and increased to 500 cells/μL in 2014 [31]. For this analysis, we included participants who were 18 years and older and excluded those enrolled in the Industrial Area of Nairobi due to small numbers of participants and differences in socioeconomic characteristics compared to Ngong Road participants [29]. Study size was limited by the number of eligible participants enrolled in the RCT.

At enrollment, participants completed a baseline questionnaire and a blood sample was collected. The baseline questionnaires collected sociodemographic, economic, and health information. Participants were randomized at enrollment, prior to ART initiation, to either SOC non-nucleoside reverse transcriptase inhibitor (NNRTI)-based ART, or were tested for PDR using an OLA to inform their initial ART regimen. The OLA is point mutation test designed to detect ≥2% mutant-frequency in a participant's HIV-quasispecies at *pol* codons K103N, Y181C, G190A, M184V, and K65R [28, 29, 32–34]. PDR was defined as having mutations detected by OLA. To prevent false-positives, low-level mutations <25% of an individual's HIV quasispecies were confirmed using Illumina sequencing described elsewhere [29]. Mutations detected by OLA but not confirmed via Illumina were defined as wild type. Those in the OLA arm with ≥10% drug resistance detected were initiated on protease inhibitor (PI)-based treatment recommended for second-line ART. ART initiation began at the first follow-up study visit scheduled approximately two weeks from enrollment. Baseline samples from participants randomized to the SOC arm were later tested for PDR and results were available to all participants at their exit visit at 15 months. Participants who missed a visit and did not respond to

several phone call attempts, received a home visit by a trained community health worker to ascertain their status and attempt to re-engage them in the study and treatment. Dates and causes of illnesses, hospitalizations, and deaths were obtained during follow-up from medical records and/or verbal autopsy via a patient's relative or other contact when available. Bias was minimized by using a prospective longitudinal study design with frequent study visits and robust follow-up methods including home visits to maximize retention and assess vital status for participants who missed visits.

## Statistical analyses

Baseline sociodemographic, economic, and health characteristics among adult enrolled patients seeking ART initiation were described for the cohort overall and compared by clinic site (Nairobi vs. Maseno) to assess differences by location using a t-test assuming unequal variance for continuous variables and a Chi-square test for binary and categorical variables. Correlates associated with not initiating ART were assessed by logistic regression to understand difference between enrolled, ART eligible, participants who did and did not attend the ART initiation visit due to known death, withdrawing from the study, or loss to follow-up.

We compared mortality incidence rates among patients who attended their first follow-up visit to initiate ART, from ART initiation visit to death date. Participants who initiated ART but withdrew from the study or were lost to follow-up were censored at the date of their last attended visit and those who completed follow-up were censored at 365 days after ART initiation. Participants who transferred to a different clinic location were censored at the date of their last visit attended at the clinic at which they enrolled. Deaths caused by unexpected injuries (e.g. motor vehicle accidents), rather than illnesses, were excluded as outcomes and these individuals were censored at their date of death. Deaths with unknown causes were included as outcomes.

Potential correlates investigated included location (Maseno vs. Nairobi), age group (18–24, 25–34, 35–49, $\geq$50), gender (male vs. female), relationship status (married or attached vs. single), years of education (0–11 vs. $\geq$12), employment status (unemployed vs. employed), sanitation access (flush toilet vs. pit latrine), and travel time to clinic (continuous). Unemployment may be associated with or caused by illness associated with mortality in addition to socioeconomic status, so was excluded from multivariable analyses due to issues of collinearity. We also investigated mortality risk by baseline health indicators including standard BMI categories ($<18.5$ m/kg$^2$ [underweight], 18.5–24.9 m/kg$^2$ [healthy], $\geq$25 m/kg$^2$ [overweight/obese]), CD4 lymphocyte count categories defined by commonly used ranges ($<100$, 100–199, 200–349, $\geq$350 cells/$\mu$L), and PDR (vs. wild-type). To investigate the potential impact of the RCT intervention, we compared mortality among those with $\geq$10% PDR detected at enrollment (randomized to receive resistance-guided-treatment) by study arm. Cox proportional hazards regression with robust standard errors was used to compare mortality risk by these potential correlates in unadjusted analyses. To investigate the independent relationship between these variables and mortality, we adjusted for combinations of likely correlates in multivariable Cox proportional hazards regression models. Correlates associated with mortality at P$\leq$0.05 in unadjusted regression and those selected a priori as likely mortality correlates were included in the multivariable models. CD4 count and BMI were excluded from the initial multivariable model to investigate correlations between sociodemographic variables and mortality when not adjusting for these strong clinical predictors. CD4 count and BMI were then included separately in subsequent multivariable models to account for collinearity between these variables and determine independent effects of sociodemographic variables, and finally included together to assess independent effects of all potential correlates. Age and years of education were included as continuous variables in all multivariable regression models. We also stratified

univariable and multivariable analyses by location to investigate differences in mortality corre-
lates and risk between Maseno (rural) and Nairobi (urban). Cox proportional hazards regres-
sion, enables us to control for losses to follow-up and minimize biases in our analyses. Those
with missing data were excluded from the regression analyses in which those variables were
included.

Kaplan-Meier survival curves show survival from ART initiation visit by select correlates
identified in regression. Curves were stratified by location for correlates with an association
that differed by clinic site.

## Results

### Participant characteristics

Descriptive statistics on demographics, socioeconomics, and baseline health and laboratory infor-
mation are shown for the 867 adults enrolled overall and by clinic location among 655 partici-
pants at the Nairobi (Ngong Road) clinic, and 212 at the Maseno clinic (Table 1). Age was similar
between clinics, with a median of 38 years. More women enrolled in Maseno than Nairobi (73%
vs. 64%; $P<0.05$). Nairobi participants had greater median number of years of education com-
pared to Maseno (12 vs. 8 years; $P<0.001$). More participants in Maseno were unemployed than
in Nairobi (38% vs. 14%; $P<0.001$) and fewer had access to a flush toilet (6% vs. 61%; $P<0.001$).
Cost of and time spent traveling to the clinic were slightly greater in Nairobi ($P<0.05$). More par-
ticipants were underweight (BMI $<18.5$ kg/m$^2$) in Maseno than Nairobi (28% vs. 13%; $P<0.001$).
More participants in Nairobi had a CD4 cell count $<50$ cells/μL than in Maseno (16% vs. 9%;
$P<0.05$), and fewer $\geq350$ cells/μL (12% vs. 18%; $P<0.05$). Slightly more participants in Maseno
than Nairobi had PDR (12% vs. 9%) but this was not statistically significant.

### Enrollment, ART initiation, and follow-up summary

Of the 867 enrolled participants, 20 (2%) were known to have died and 28 (3%) withdrew,
transferred, or were lost to follow-up prior to initiating ART. Overall, 612 (93%) in Nairobi
and 207 (98%) in Maseno initiated ART. Of those who initiated ART, 56 (7%) died (including
1 auto accident), 52 (6%) withdrew or were lost to follow-up, and 8 (1%) transferred clinics
within 12 months (Fig 1). Causes and/or symptoms reported at time of death are described (S1
Table). Those who did not initiate ART (n = 48) were more likely to be in Nairobi ($P = 0.026$),
unemployed ($P = 0.001$), and have CD4 count $<100$ cells/μL ($P = 0.035$), compared to those
who initiated ART (Table 2); among these, 20 (42%) were known mortalities and the remain-
ing 28 were lost to follow-up with unknown vital status.

Among those who initiated ART, the average time from enrollment to ART initiation was
21 days (median 16; IQR: 16–22) overall, 22 days in Nairobi (median 17 days; IQR: 14–23),
and 18 days in Maseno (median 14 days; IQR: 14–21). The average follow-up time within 365
days from ART initiation was 330 days overall, 334 in Nairobi, and 317 in Maseno (overall and
by location the medians were 365 days and the IQRs were 365–365). Among those who were
not reported dead within 365 days from ART initiation, the average follow-up time was 346
days overall, 347 in Nairobi, and 346 in Maseno (overall and by location the medians were 365
days and the IQRs were 365–365).

### Mortality incidence and correlates of mortality risk following ART
### initiation

Of the participants who initiated ART, 55 (7%) died from a non-injury related cause within
365 days of ART initiation, including 32 (5%) in Nairobi and 23 (11%) in Maseno. The median

**Table 1. Characteristics of enrolled adult participants eligible to initiate ART by clinic location.**

| Characteristics[a] | Urban Nairobi (n = 655) | Rural Maseno (n = 212) | Total (n = 867) |
|---|---|---|---|
| **Demographic** | | | |
| Age in years | 38 (32, 45) | 39 (30, 47) | 38 (31, 46) |
| Female | 421 (64%) | 155 (73%)* | 576 (66%) |
| **Socioeconomic** | | | |
| Married/steady partner | 396 (60%) | 135 (64%) | 531 (61%) |
| Education in years[b] | 12 (8, 14) | 8 (7, 10)** | 11 (8, 13) |
| Unemployed[b] | 89 (14%) | 80 (38%)** | 169 (19%) |
| Monthly rent in US$[b] | 39 (0, 89) | 0 (0, 0)** | 22 (0, 66) |
| Flush toilet[b] | 396 (61%) | 12 (6%)** | 408 (47%) |
| Persons living in house | 3 (2, 5) | 4 (3, 5)** | 4 (2, 5) |
| **Access to Care** | | | |
| Cost of travel in US$[b] | 2.22 (1.11, 2.77) | 2.10 (1.11, 3.32)* | 2.22 (1.11, 2.88) |
| Travel time to clinic in hours | 1 (0.67, 2.00) | 1 (0.50, 1.50)* | 1 (0.67, 2.00) |
| **Health & Laboratory (at Baseline)** | | | |
| BMI (kg/m$^2$) | 23 (20, 26) | 21 (18, 23)** | 22 (19, 25) |
| <18.5 (underweight) | 85 (13%) | 60 (28%)** | 158 (18%) |
| 18.5–24.9 (healthy) | 355 (56%) | 114 (54%) | 468 (54%) |
| 25–29.9 (overweight) | 136 (21%) | 34 (16%) | 170 (20%) |
| ≥30 (obese) | 59 (9%) | 4 (2%)** | 68 (8%) |
| **CD4 count (cells/μL)[b]** | 224 (97, 305) | 233 (135, 323) | 227 (105, 308) |
| <50 | 102 (16%) | 19 (9%)* | 121 (14%) |
| 50–99 | 63 (10%) | 22 (10%) | 85 (10%) |
| 100–199 | 135 (21%) | 43 (20%) | 178 (21%) |
| 200–349 | 276 (42%) | 88 (42%) | 364 (42%) |
| 350–499 | 63 (10%) | 39 (18%)** | 102 (12%) |
| ≥500 | 14 (2%) | 0 (0%)* | 14 (2%) |
| Viral load (log10, copies/mL)[b] | 4.75 (4.08, 5.30) | 4.41 (3.76, 5.12)** | 4.67 (3.97, 5.23) |
| Drug resistance ≥2%, OLA[b, c] | 58 (9%) | 25 (12%) | 83 (10%) |
| Drug resistance ≥10%, OLA[b, c] | 50 (8%) | 19 (9%) | 69 (8%) |
| **Study Intervention and ART initiation** | | | |
| Randomized at enrollment to OLA informed ART | 329 (50%) | 112 (53%) | 441 (51%) |
| Randomized at enrollment to OLA & had drug resistance ≥10% | 30 (5%) | 11 (5%) | 41 (5%) |
| ART initiation visit attended | 612 (93%) | 207 (98%)* | 819 (94%) |

Abbreviations: ART, Antiretroviral therapy; OLA, oligonucleotide ligation assay (point mutation test designed to detect K103N, Y181C, M184V, G190A, and K65R)

[a] For continuous variables, median (interquartile range) are presented. For binary and categorical variables, the number (%) within that category is shown.

[b] Data is complete except for the following variables for Nairobi: Monthly rent (n = 633), Type of toilet (n = 654), Cost of travel (n = 642), BMI (n = 635), CD4 count (n = 653), Viral load (n = 548), Drug resistance (n = 652); Data is complete for Maseno except for Viral load (n = 177). Viral load testing was not performed for participants who completed fewer than 4 months of follow-up.

[c] OLA is a point mutation test designed to detect K103N, Y181C, M184V, G190A, and K65R. Percent resistant is defined by the highest frequency of viral variant with a mutant codon detected within an individual's HIV-quasispecies.

T-test assuming unequal variance for continuous variables and a chi$^2$ test for binary and categorical variables used to compare across locations. For BMI and CD4 count categories, proportions are compared within each category across locations with chi$^2$ test.

*p<0.05,

**p<0.001

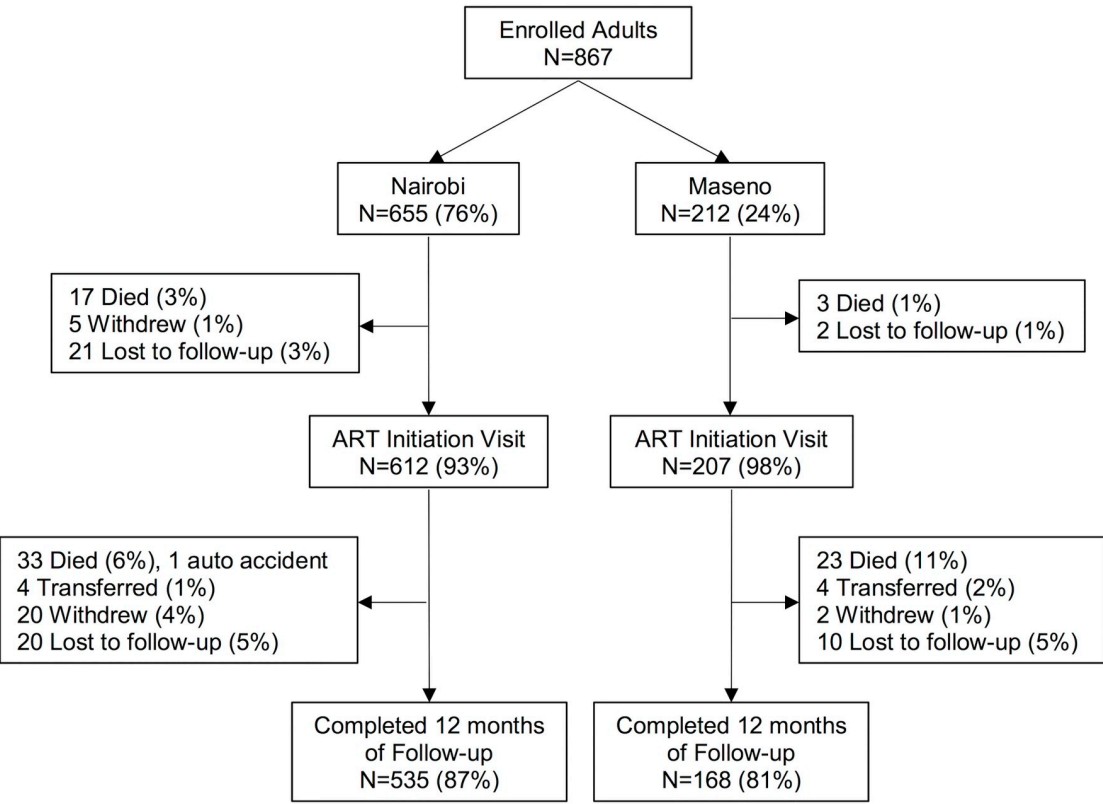

**Fig 1. Flow chart from enrollment of adult participants.** Flow chart diagramming overall study follow-up and attrition before and after ART initiation by location (Nairobi and Maseno).

time to death from ART initiation was 64 days (IQR: 24–152) overall, 69 days (IQR: 25–132) in Nairobi, and 62 days (IQR: 24–152) in Maseno. Overall, of those who died within a year from initiating ART, 18 (33%), 25 (45%), 37 (67%), and 44 (80%) died within 30, 60, 90, and 180 days from ART initiation. The overall mortality incidence rate within a year of initiating ART was 7.44 per 100 person-years (95% CI 5.71, 9.69).

In unadjusted Cox proportional hazards regression, the Maseno location, older age, male gender, fewer years of education, unemployment, low CD4 count, low BMI, and PDR were associated with increased mortality risk within a year of ART initiation (Table 3; Fig 2). Increased risk of mortality associated with age (HR 1.04 for a one-year increase; 95% CI 1.02, 1.07; $P<0.001$) persisted in models adjusted for location, gender, education, PDR, CD4 count, and BMI (Table 4). Males had 1.74-fold increased risk of mortality than females (95% CI 1.02–2.95; $P = 0.041$), which remained when adjusting for location, age, education and PDR, but not when adjusting for BMI and/or CD4 count. A one-year increase in education was associated with a decreased risk of mortality (HR 0.92; 95% CI 0.88, 0.97; $P = 0.002$), which remained when adjusting for other variables. Unemployment was associated with an increased risk in unadjusted analyses (HR 1.89; 95% CI 1.05, 3.40; $P = 0.033$). Participants with a CD4 count $<100$ had a 11.67-fold increased risk of mortality compared to those with 200–349 cells/μL (95% CI 4.93, 27.65; $P<0.001$). Participants with a low BMI ($<18.5$ m/kg$^2$) vs. healthy BMI (18.5–24.9 m/kg$^2$) had a 4.99-fold increased risk (95% CI 2.79, 8.92; $P<0.001$). The associations between CD4 and BMI with increased mortality risk persisted in multivariable analyses. Those with PDR ($\geq$2% detected via OLA) had a 2.49-fold increased risk of mortality than

**Table 2. Correlates of enrollees not returning to study to initiate ART[*].**

| Variable | | Odds ratio for not initiating ART |
|---|---|---|
| Clinic Location | Nairobi | Ref |
| | Maseno | 0.34 (0.13, 0.88); **0.026** |
| Age | Continuous (1-year increase) | 0.98 (0.96, 1.01); 0.192 |
| Gender | Female | Ref |
| | Male | 1.32 (0.73, 2.39); 0.365 |
| Marital Status | Single | Ref |
| | Married/Attached | 0.88 (0.49, 1.59); 0.670 |
| Education | Continuous (1-year increase) | 0.99 (0.92, 1.06); 0.721 |
| Employment Status | Employed | Ref |
| | Unemployed | 2.92 (1.60, 5.35); **0.001** |
| Type of Toilet | Pit Latrine | Ref |
| | Flush | 1.35 (0.75, 2.42); 0.316 |
| Travel time to clinic | Continuous (1-hour increase) | 1.00 (0.92, 1.08); 0.924 |
| BMI (kg/m$^2$) | <18.5 (underweight) | 1.61 (0.81, 3.21); 0.175 |
| | 18.5–24.9 (healthy) | Ref |
| | ≥25 (overweight/obese) | 0.51 (0.22, 1.18); 0.116 |
| CD4 Count (cells/μL) | <100 | 1.95 (1.05, 3.63); **0.035** |
| | ≥100 | Ref |
| Pre-Treatment Drug Resistance | Wild-type (no PDR) | Ref |
| | PDR ≥2% | 1.48 (0.61, 3.62); 0.387 |
| Study randomization | SOC arm | Ref |
| | OLA arm | 1.05 (0.59, 1.89); 0.862 |

[*]Unadjusted logistic regression for death, withdraw, or lost to follow-up prior to ART start visit (95% confidence intervals); p-value. P-values<0.05 are in bold.

those with wild-type virus (95% CI 1.29–4.79; *P* = 0.006), which remained when adjusting for location, age, gender, education, and BMI, but not when adjusting for CD4 count. There was no statistically significant difference in mortality risk between those who did or did not receive the RCT intervention. There was no significant association for relationship status and mortality risk. Sanitation (type of toilet) was collinear with location (see Table 1), so was excluded from this analysis.

Maseno had a 2.20-fold greater risk of mortality than Nairobi (95% CI 1.29, 3.76; *P* = 0.004) (Table 3; Fig 2). This association remained when adjusting for age, gender, education, PDR, and CD4 count, but not when adjusting for BMI (Table 4). When stratifying by location (Table 5) we found CD4 count and BMI were associated with mortality at both locations, while older age and male gender were only statistically significantly associated with mortality in Nairobi. PDR was only associated with mortality in Maseno. When adjusting for the other variables, the association between CD4 count and BMI remained for both sites, as did older age and male gender for Nairobi, and PDR for Maseno. Lower education in Nairobi, and age and female gender in Maseno were associated with mortality in adjusted stratified analyses. The association between CD4 count, BMI, and PDR with increased mortality risk was greater in Maseno than in Nairobi in unadjusted analyses (Fig 3). Adjusted associations between mortality and CD4 count and PDR remained greater in Maseno than Nairobi, though were similar across locations for BMI; only effect modification by location for CD4 count was statistically significant (*P*<0.001).

**Table 3. Unadjusted incidence rates and hazard ratios (HR) of mortality following ART initiation (N = 811)[a].**

| Variables | | Deaths/person-years | Incidence (95% CI)[b] | HR (95% CI); p-value[c] |
|---|---|---|---|---|
| Overall | - | 55/779 | 7.44 (5.71, 9.69) | - |
| Location | Nairobi | 32/559 | 5.72 (4.05, 8.09) | Ref |
| | Maseno | 23/180 | 12.78 (8.49, 19.23) | 2.20 (1.29, 3.76); **0.004** |
| Age | 18–24 | 0/35 | 0 | Ref[e] |
| | 25–34 | 16/238 | 6.71 (4.11, 10.95) | |
| | 35–49 | 21/349 | 6.02 (3.93, 9.24) | 1.03 (0.54, 1.98); 0.918 |
| | ≥50 | 18/117 | 15.35 (9.67, 24.37) | 2.59 (1.33, 5.05); **0.005** |
| | 1-year increase | - | - | 1.04 (1.02, 1.07); <**0.001** |
| Gender | Female | 30/502 | 5.97 (4.17, 8.54) | Ref |
| | Male | 25/237 | 10.60 (7.14, 15.63) | 1.74 (1.02, 2.95); **0.041** |
| Relationship Status | Single | 20/284 | 7.04 (4.54, 10.92) | Ref |
| | Married/attached | 35/455 | 7.69 (5.52, 10.71) | 1.09 (0.63, 1.88); 0.766 |
| Education Years | 0–11 | 36/379 | 9.49 (6.85, 13.16) | Ref |
| | ≥12 | 19/360 | 5.28 (3.37, 8.28) | 0.56 (0.32, 0.98); **0.042** |
| | 1-year increase | - | - | 0.92 (0.88, 0.97); **0.002** |
| Employment Status | Employed | 39/609 | 6.40 (4.68, 8.76) | Ref |
| | Unemployed | 16/130 | 12.32 (7.55, 20.12) | 1.89 (1.05, 3.40); **0.033** |
| BMI Category (m/kg$^2$) | <18.5 (underweight) | 27/103 | 26.25 (18.00, 38.28) | 4.99 (2.79, 8.92), <**0.001** |
| | 18.5–24.9 (healthy) | 20/405 | 4.94 (3.19, 7.65) | Ref |
| | ≥25 (overweight/obese) | 6/215 | 2.79 (1.25, 6.22) | 0.57 (0.23, 1.41); 0.224 |
| CD4 Count (cells/μL) | <100 | 35/156 | 22.40 (16.08, 31.20)[b] | 11.67 (4.93, 27.65); <**0.001** |
| | 100–199 | 10/157 | 6.35 (3.42, 11.81) | 3.40 (1.24, 9.34); **0.018** |
| | 200–349 | 6/326 | 1.84 (0.83, 4.09) | Ref |
| | ≥350 | 3/99 | 3.04 (0.98, 9.41) | 1.63 (0.41, 6.47); 0.491 |
| PDR | 0% (wild-type) | 44/673 | 6.54 (4.86, 8.78) | Ref |
| | ≥2% | 11/66 | 16.68 (9.24, 30.12) | 2.49 (1.29, 4.79); **0.006** |
| | 2–9% | 4/9 | 43.87 (16.47, 116.90) | 6.17 (2.44, 15.59); <**0.001** |
| | 10–100%[d] | 7/57 | 12.32 (5.87, 25.84) | 1.86 (0.83, 4.13); 0.129 |
| Intervention | PDR 10–100%, OLA arm | 5/32 | 15.49 (6.45, 37.21) | Ref |
| | PDR 10–100%, SOC arm | 2/25 | 8.15 (2.04, 32.58) | 0.54 (0.10, 2.80); 0.462 |

Abbreviations: ART, Antiretroviral therapy; HR, Hazard ratio; CI, confidence interval; BMI, Body mass index; PDR, Pre-treatment drug resistance; OLA, Oligonucleotide ligation assay; SOC, Standard of care; Ref, reference category.

[a]See Footnote in Table 1 for information on missing variable information.

[b]Incidence per 100 person-years.

[c]HRs estimated using Cox proportional hazards regression with robust variance estimates. P-values<0.05 are in bold.

[d]Approximately 50% of these individuals were randomized to OLA testing for PDR, and those with ≥10% drug resistant variants in their HIV-quasispecies were initiated on protease-inhibitor-based ART (which was shown to reduce their rate of virologic failure (submitted))

[e]The reference group for age is 18–34 years.

## Discussion

In this study of HIV infected adults in Kenya in 2013/14, we estimated the risk and identified correlates of mortality within a year of ART initiation. Overall, 7% of participants were known to have died within a year of initiating ART. This is similar to the 9% incidence estimated in a 2011 meta-analysis of studies from sub-Saharan Africa [9]. Compared to a large study of patients in Europe and North America [35], the mortality rates within a year were an order

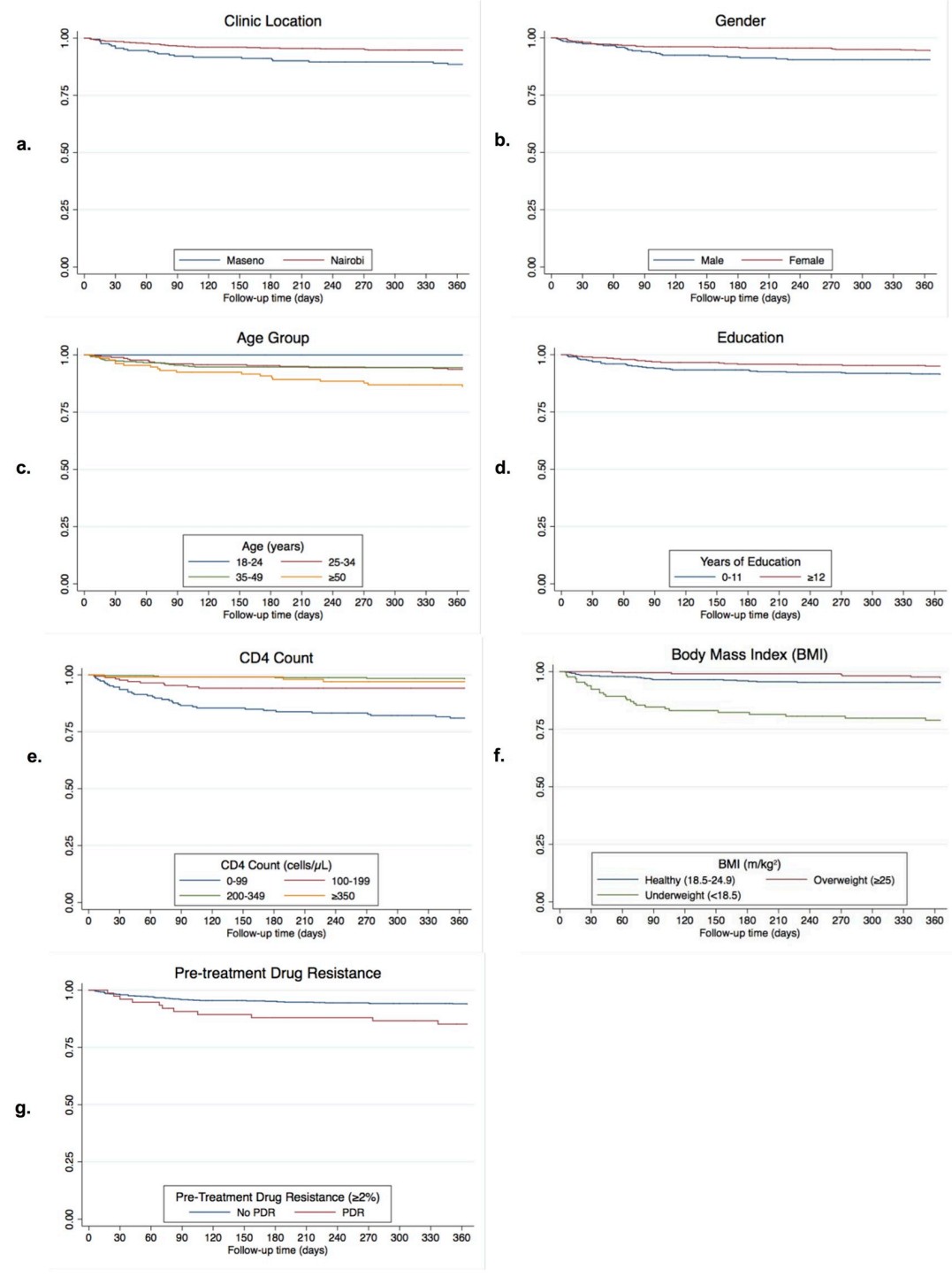

**Fig 2. Kaplan-Meier curves from ART initiation to death by correlates of mortality.** Kaplan-Meier survival curves from ART initiation to death illustrating survival by correlates of mortality in the combined cohort by a) location, b) gender, c) age group, d) education, e) CD4 count, f) body mass index (BMI), and g) pre-treatment drug resistance (PDR).

**Table 4. Adjusted hazard ratios (HR) of mortality following ART initiation (N = 811)[a].**

| Variables | Model 1 (N = 811) HR (95% CI); p-value[b] | Model 2 (N = 810) HR (95% CI); p-value[b] | Model 3 (N = 792) HR (95% CI); p-value[b] | Model 4 (N = 791) HR (95% CI); p-value[b] |
|---|---|---|---|---|
| Maseno vs. Nairobi | 1.84 (1.06, 3.19); **0.029** | 2.09 (1.17, 3.74); **0.013** | 1.31 (0.73, 2.33); 0.364 | 1.55 (0.82, 2.95); 0.181 |
| Age (1-year increased) | 1.04 (1.01, 1.06); **0.003** | 1.03 (1.01, 1.05); **0.002** | 1.04 (1.02, 1.06); **<0.001** | 1.03 (1.01, 1.05); **<0.001** |
| Male vs. female | 1.79 (1.02, 3.13); **0.041** | 1.18 (0.63, 2.20); 0.606 | 1.43 (0.80, 2.55); 0.233 | 1.11 (0.59, 2.10); 0.747 |
| Education (1-year increased) | 0.95 (0.90, 1.00); **0.049** | 0.93 (0.88, 0.98); **0.008** | 0.96 (0.91, 1.01); 0.101 | 0.93 (0.89, 0.98); **0.010** |
| PDR ≥2% | 2.76 (1.43. 5.32); **0.002** | 1.69 (0.90, 3.21); 0.105 | 2.49 (1.32, 4.68); **0.005** | 1.46 (0.75, 2.86); 0.266 |
| CD4 count category (cells/μL) | | | | |
| <100 | | 11.37 (4.72, 27.39); **<0.001** | | 7.97 (3.20, 19.87); **<0.001** |
| 100–199 | - | 3.53 (1.29, 9.65); **0.014** | - | 2.82 (1.01, 7.88); **0.049** |
| 200–349 | | Ref | | Ref |
| ≥350 | | 1.28 (0.30, 5.48); 0.735 | | 1.39 (0.33, 5.94); 0.653 |
| BMI Category (m/kg²) | | | | |
| <18.5 (underweight) | | | 4.41 (2.51, 7.75); **<0.001** | 3.11 (1.69, 5.74); **<0.001** |
| 18.5–24.9 (healthy) | - | - | Ref | Ref |
| ≥25 (overweight/obese) | | | 0.59 (0.23, 1.47); 0.257 | 0.87 (0.34, 2.24); 0.779 |

Abbreviations: ART, Antiretroviral therapy; HR, Hazard ratio; CI, confidence interval; BMI, Body mass index; PDR, Pre-treatment drug resistance; OLA, Oligonucleotide ligation assay; SOC, Standard of care; Ref, reference category.

[a]Of those who initiated ART, CD4 count was missing for 1 participant and BMI information was missing for 19 participants.

[b]HRs were estimated using Cox proportional hazards regression with robust variance estimates. For each model, we adjusted for all variables with results presented. P-values<0.05 are in bold.

of magnitude higher in our study for those with a low CD4 count <100, but similar at CD4 counts >200 cells/μL. The majority (67%) of deaths in our study occurred within 3 months of initiating ART. This elevated risk of mortality within the first few months of ART initiation is consistent with other studies in sub-Saharan Africa and globally [13, 35–38]. Interventions to modify the risk of early mortality may be most effective by targeting this time-frame, in addition to efforts to diagnose and treat individuals earlier in HIV disease progression.

We found that a low CD4 lymphocyte count, low BMI, rural location, increased age, male gender, fewer years of education, unemployment, and PDR were associated with greater risk of mortality. Low CD4, low BMI, and PDR were associated with a greater risk of mortality at the rural location compared to those at the urban location. Because the clinics were designed and managed by the Coptic Hospital to provide the same high level of services and programs [26, 27], differences by location are more likely due to regional or rural/urban disparities in underlying health and infectious disease burden [19]. The higher risk of death in rural Maseno compared to urban Nairobi remained even when controlling for CD4 count, but not when controlling for BMI indicating that poor nutrition may explain some of the higher risk of mortality in this rural setting. Stratified analyses suggest that the consequences of poor nutrition, low pre-ART CD4 count, and drug resistance may be more severe in rural settings where the risk of coinfections is higher [22–25]. Providing ARV-naïve individuals with point-of-use water filtration and/or long-lasting insecticide-treated bed nets has been shown to prevent diarrheal disease and malaria and delay HIV disease progression [39, 40]. While evidence is needed to determine if such interventions would be effective at reducing short-term mortality among individuals with more advanced HIV progression initiating ART, more aggressive management of coinfections has been shown to be beneficial in the REALITY trial and could improve outcomes for late presenters [41].

**Table 5. Univariable and multivariable Cox proportional hazards regression for mortality from ART initiation visit by location (N = 811)[a].**

| Variable | Nairobi (N = 606) | | Maseno (N = 205) | |
|---|---|---|---|---|
| | Unadjusted HR[b] | Adjusted HR[b] | Unadjusted HR[b] | Adjusted HR[b] |
| Age (1yr increase) | 1.05 (1.02, 1.08); **0.003** | 1.05 (1.01, 1.08); **0.009** | 1.03 (1.00, 1.06); 0.062 | 1.04 (1.01, 1.07); **0.002** |
| Male vs. Female | 2.16 (1.08, 4.31); **0.030** | 2.21 (1.01, 4.82); **0.047** | 1.50 (0.64, 3.49); 0.348 | 0.26 (0.10, 0.64); **0.003** |
| Married/Attached | 1.24 (0.60, 2.57); 0.566 | - | 0.85 (0.37, 1.95); 0.695 | - |
| School years (1yr increase) | 0.94 (0.88, 1.00); 0.051 | 0.90 (0.83, 0.97); **0.005** | 0.96 (0.87, 1.05); 0.345 | 0.91 (0.82, 1.02); 0.095 |
| Unemployed | 1.76 (0.72, 4.26); 0.213 | - | 1.32 (0.58, 3.02); 0.510 | - |
| Flush toilet vs. pit | 0.83 (0.41, 1.66); 0.597 | - | - | - |
| Time to clinic (1min increase) | 0.88 (0.64, 1.20); 0.407 | - | 1.12 (0.76, 1.67); 0.561 | - |
| PDR ≥2% | 1.55 (0.55, 4.40); 0.408 | 0.63 (0.17, 2.35); 0.495 | 3.41 (1.43, 8.16); **0.006** | 3.46 (1.62, 7.40); **0.001** |
| PDR 10–100%, OLA arm | Ref | | Ref | |
| PDR 10–100%, SOC arm | 0.69 (0.06, 7.72); 0.764 | - | 0.42 (0.05, 3.77); 0.437 | - |
| CD4 count category | | | | |
| <100 | 7.01 (2.79, 17.59); <**0.001** | 5.30 (1.90, 14.84); **0.001** | 21.64 (6.33, 74.00); <**0.001** | 20.53 (4.68, 89.98); <**0.001** |
| 100–199 | 3.00 (1.01, 8.92); **0.047** | 2.08 (0.62, 6.95); 0.234 | 2.94 (0.58, 14.80); 0.191 | 4.36 (0.81, 23.50); 0.087 |
| ≥200 | Ref | Ref | Ref | Ref |
| BMI Category | | | | |
| <18.5 (underweight) | 3.15 (1.41, 7.03); **0.005** | 3.62 (1.47, 8.90); **0.005** | 7.59 (2.79, 20.62); <**0.001** | 3.57 (1.11, 11.52); **0.033** |
| 18.5–24.9 (healthy) | Ref | Ref | Ref | Ref |
| ≥25 (overweight/obese) | 0.57 (0.21, 1.55); 0.270 | 0.93 (0.32, 2.70); 0.889 | 0.58 (0.07, 4.84); 0.614 | 0.35 (0.06, 2.18); 0.262 |

Abbreviations: ART, Antiretroviral therapy; HR, Hazard ratio; CI, confidence interval; BMI, Body mass index; PDR, Pre-treatment drug resistance; OLA, Oligonucleotide ligation assay; SOC, Standard of care; Ref, reference category.

[a]See footnote in Table 1 for information on missing variables. For adjusted models, N = 586 for Nairobi & N = 205 for Maseno.

[b]HRs estimated using Cox proportional hazards regression with robust variance estimates. Adjusted HR controls for all other variables with results presented. P-values <0.05 are in bold.

Our results are generally consistent with previous studies investigating post-ART mortality among HIV-infected adults in sub-Saharan Africa. Similar to other studies, older age was associated with mortality [7, 9, 10] and is consistent with older adults being diagnosed and presenting for treatment later, with less immune recovery during treatment [42]. Male gender has been associated with higher post-ART mortality in many studies [7, 9–11] including those conducted in coastal and Western Kenya [14, 43]. We previously found males to be at higher risk of attrition from clinic attendance at the same Coptic Hope Center in Nairobi [44]. HIV-infected men have been shown to have later diagnoses and ART initiation, worse engagement, poorer adherence, and more severe outcomes including mortality than women throughout low- and middle-income countries [11, 45]. The results of our study add to the expanding body of literature demonstrating high mortality risk among HIV-infected men and underscore the continued need to engage and retain men in care.

The independent association we found between low BMI and mortality is also consistent with prior studies [9, 46–49]. Even among ARV-naïve patients with less advanced HIV (CD4 ≥350 cellsμ/L), low BMI was associated with increased mortality risk in a study in Uganda [50]. Weight loss was found to be associated with mortality in studies of patients initiating [36] or currently on ART [51] and weight gain is associated with greater survival [49, 52, 53]. While nutritional supplementation and food assistance have effectively increased BMI in some [54–57], but not all studies [58], such interventions have not been shown to significantly decrease short-term mortality risk in HIV-infected adults [58, 59]. However, evidence is limited and

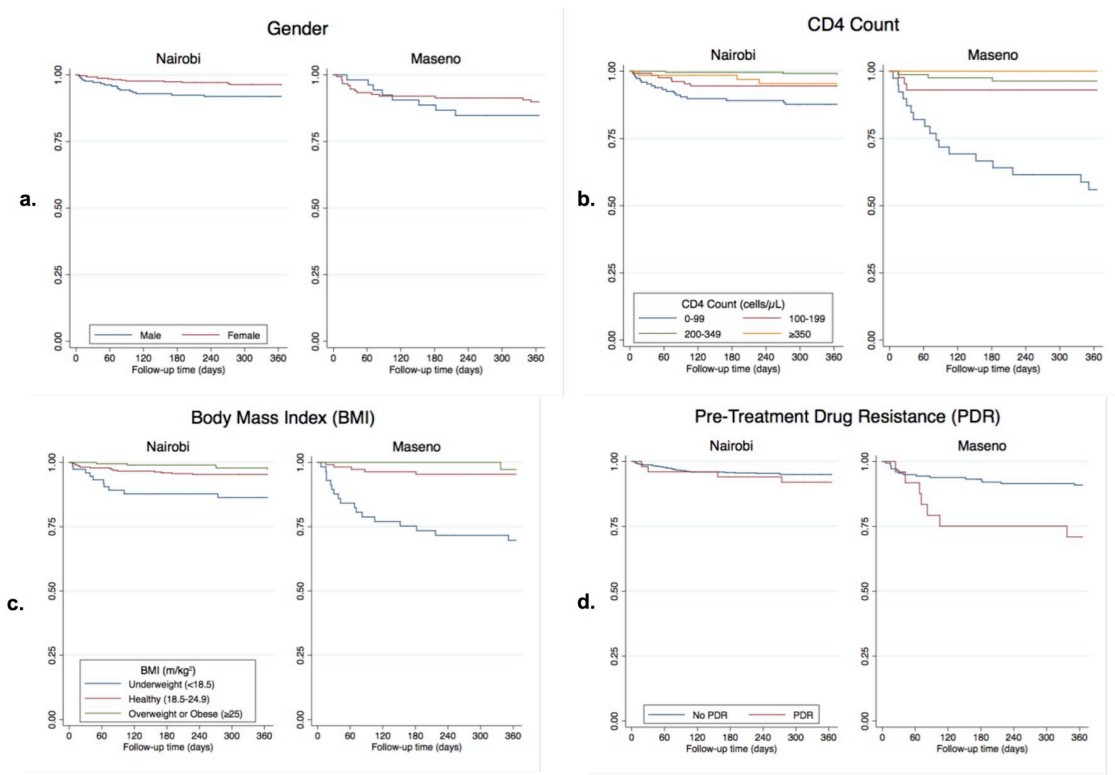

**Fig 3. Kaplan-Meier survival curves from ART initiation to death by correlates of mortality, stratified by clinic location.**
Kaplan-Meier survival curves from ART initiation to death, stratified by clinic location (Nairobi and Maseno), illustrating survival
by correlates of mortality that differed in their association with mortality by location including a) gender, b) CD4 count, c) BMI, and
d) PDR.

nutritional supplementation has been shown to be cost-effective for reducing mortality in
severely underweight individuals [60].

There is limited evidence regarding PDR and short-term mortality risk in published studies.
While PDR was not statistically significantly associated with mortality within one and two
years of ART initiation in a study across Kenya, Nigeria, South Africa, Uganda, Zambia, and
Zimbabwe [18], it was found to be associated with death among those on ART for at least 6
months in one study conducted in Malawi, Kenya, Uganda, and Cambodia [17]. In adjusted
analysis in our study, the association between PDR and mortality remained statistically signifi-
cant only among rural Maseno participants. Further study is needed to understand the mecha-
nisms by which PDR contributes to early morality after ART initiation. Given the substantial
evidence of virologic failure and poor health outcomes among patients with PDR initiating
ART in resource limited settings [17, 34, 61, 62] and observed increases in PDR prevalence
[28, 63], scale-up of resistance testing and/or alternative ARV combinations may be warranted.
Utilizing ARVs like dolutegravir with a higher barrier of resistance [64] could be beneficial
in Kenya and similar settings where first-line regimen recommendations currently include
NNRTI based ART [2, 65].

There is mixed evidence regarding the association between socioeconomic status and
short-term mortality among HIV-infected individuals initiating ART [12–15]. We found that
greater years of education and employment were protective, and unemployment was also asso-
ciated with not initiating ART (many non-initiators were known mortalities). While unem-
ployment may be associated with underlying severe illness leading to both inability to work

and early mortality, the independent association found with education suggests less educated individuals may require additional support to mitigate their higher risk of short-term mortality. There is mixed evidence that single marital status may be associated with higher risk of HIV-related mortality [13, 14], and we did not observe an association in our study.

Study limitations include that baseline viral loads were not determined on all subjects who died or were lost to follow-up, so could not be used in regression analyses, and use of a single pre-enrollment CD4 count measurement [66, 67]. However, CD4 count has commonly been used in clinical settings to define the health and severity of HIV-infected individuals [2, 6, 35, 68, 69]. More direct measures of socioeconomic status, like income, were unavailable for our analyses. Our study also did not investigate the impact of poor adherence to medications nor quantify non-fatal indicators of poor health. Data to specifically identify immune reconstitution inflammatory syndrome (IRIS) were not collected, though the timing of most deaths suggests that looking for IRIS may be an important intervention. The results of our study may not be generalizable to the HIV infected population in Kenya given the intensity of study follow-up often not feasible for patients in a normal clinic setting, and that study data represents only two clinics located in separate geographic regions. Although our study was nested in an RCT, 82.7% of screened participants were enrolled [29] suggesting reasonable coverage of the population in care. We also found no significant difference in mortality risk due to the RCT intervention. Our study has notable strengths as a large prospective cohort study with careful follow-up and tracking, assessment of mortality, and high retention. For example, only 5% of participants were lost to follow-up within 12 months in our study, which is much lower than the 20% within 6 months and 10% within 6–12 months of initiating ART reported in the large cohort in Kenya within the International Epidemiologic Databases to Evaluate AIDS (IeDEA) Collaboration [8]. High retention in our study was likely due to intensive follow-up and contributes to more precise and robust mortality risk estimates in our study. The Coptic Hospital Hope Center clinics are designed to provide uniform high standard of care [26, 27] across regional locations, allowing us to look beyond health service delivery as a contributor of differences in mortality. Using a prospective longitudinal study design with monthly/bi-monthly follow-up visits, we were able to control for losses to follow-up and minimize biases in our analyses using Cox proportional hazards regression.

## Conclusions

We found a high proportion of HIV-infected patients initiating ART with low CD4 counts, indicative of delayed treatment and increased risk for poor health outcomes and transmission to others. This study identifies multiple potentially modifiable risk factors associate with increased mortality within the first year of ART. Targeted interventions to patients with a low CD4 count at presentation, as well as to those who are older, male, less educated and unemployed, and those with low BMI or PDR may help mitigate the risk of early mortality in Kenya and similar populations, especially in rural areas.

## Supporting information

**S1 Table. List of baseline correlates and summary of cause/symptoms at the time of death (n = 81).** Details on ART initiation status, gender, age, body mass index, CD4 count, pre-treatment drug resistance status, number of days from study enrollment to death, time from ART initiation to death, and summary of cause of death and/or symptoms at time of death when available by location: A) Nairobi and B) Maseno.
(DOCX)

## Acknowledgments

We thank the study participants and their families who are committed to advancing HIV care, and the research personnel, clinic and laboratory staff, and data management teams in Nairobi and Seattle for their efforts as well as the Coptic Hope Center for Infectious Diseases and its patients.

## Author Contributions

**Conceptualization:** Rachel A. Silverman, Grace C. John-Stewart, Lisa M. Frenkel, Michael H. Chung.

**Data curation:** Rachel A. Silverman, Ingrid A. Beck, Catherine Kiptinness, Christine J. McGrath, Lisa M. Frenkel, Michael H. Chung.

**Formal analysis:** Rachel A. Silverman, Christine J. McGrath.

**Funding acquisition:** Lisa M. Frenkel.

**Investigation:** Rachel A. Silverman, Ingrid A. Beck, Ross Milne, Catherine Kiptinness, Michael H. Chung.

**Methodology:** Rachel A. Silverman, Grace C. John-Stewart, Barbra A. Richardson, Lisa M. Frenkel, Michael H. Chung.

**Project administration:** Rachel A. Silverman, Ingrid A. Beck, Ross Milne, Catherine Kiptinness, Christine J. McGrath, Bhavna Chohan, Samah R. Sakr, Lisa M. Frenkel, Michael H. Chung.

**Resources:** Ingrid A. Beck, Samah R. Sakr, Lisa M. Frenkel, Michael H. Chung.

**Software:** Rachel A. Silverman.

**Supervision:** Grace C. John-Stewart, Ingrid A. Beck, Catherine Kiptinness, Bhavna Chohan, Lisa M. Frenkel, Michael H. Chung.

**Validation:** Rachel A. Silverman, Ingrid A. Beck, Christine J. McGrath, Barbra A. Richardson, Lisa M. Frenkel.

**Visualization:** Rachel A. Silverman.

**Writing – original draft:** Rachel A. Silverman.

**Writing – review & editing:** Rachel A. Silverman, Grace C. John-Stewart, Ingrid A. Beck, Ross Milne, Christine J. McGrath, Barbra A. Richardson, Samah R. Sakr, Lisa M. Frenkel, Michael H. Chung.

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
