## [Decision Letter · Decision Letter 0]

23 Jul 2019

PONE-D-19-18343

Predictors of mortality within the first year of initiating antiretroviral therapy in urban and rural Kenya

PLOS ONE

Dear Dr. Silverman,

Thank you for submitting your manuscript to PLOS ONE. After careful consideration, we feel that it has merit but does not fully meet PLOS ONE’s publication criteria as it currently stands. Therefore, we invite you to submit a revised version of the manuscript that addresses the points raised during the review process.

There are just a few comments from reviewer #2, which I would encourage you to address in a revision. In addition to those comments, I would make the following suggestions:

As this is a cohort study (nested within RCT) please ensure that reporting conforms with the STROBE guidelines (available at http://www.equator-network.org/reporting-guidelines/strobe/). Please submit checklist with revised submission.In the abstract, I think it would be more interesting to present the results of the multivariable Cox regression analysis as opposed to the unadjusted analysesI wasn't sure of your definition of PDR as an explanatory variable for the regression analyses. Was this based on OLA (i.e. just the mutations detected by OLA) or based on Illumina NGS (i.e. any drug resistance mutation)? It would be helpful to make this clearer in the Methods section.I couldn't see clear information in the Methods section about how missing data were handled in the regression analysis. It looks from the table footnotes that there may have been only few missing values for the variables included in the analysis, but it would still be helpful to explain clearly how you handled this.

We would appreciate receiving your revised manuscript by Sep 06 2019 11:59PM. To enhance the reproducibility of your results, we recommend that if applicable you deposit your laboratory protocols in protocols.io, where a protocol can be assigned its own identifier (DOI) such that it can be cited independently in the future. For instructions see: http://journals.plos.org/plosone/s/submission-guidelines#loc-laboratory-protocols

We look forward to receiving your revised manuscript.

Kind regards,

Richard John Lessells, BSc, MBChB, MRCP, DTM&H, DipHIVMed, PhD

Academic Editor

PLOS ONE

Journal Requirements:

2) We note that you have indicated that data from this study are available upon request. PLOS only allows data to be available upon request if there are legal or ethical restrictions on sharing data publicly. For information on unacceptable data access restrictions, please see http://journals.plos.org/plosone/s/data-availability#loc-unacceptable-data-access-restrictions.

Reviewers' comments:

Reviewer's Responses to Questions

**Comments to the Author**

1. Is the manuscript technically sound, and do the data support the conclusions?

Reviewer #1: Yes

Reviewer #2: Yes

2. Has the statistical analysis been performed appropriately and rigorously? 

Reviewer #1: Yes

Reviewer #2: Yes

3. Have the authors made all data underlying the findings in their manuscript fully available?

Reviewer #1: Yes

Reviewer #2: No

4. Is the manuscript presented in an intelligible fashion and written in standard English?

Reviewer #1: Yes

Reviewer #2: Yes

5. Review Comments to the Author

Reviewer #1: In my opinion the study presents the results of primary scientific research, statistics, and other analyses are performed to a high technical standard and are described in sufficient detail, conclusions are presented in an appropriate fashion and are supported by the data.

The article is presented in an intelligible fashion and is written in standard English.

In my opinion, t he research meets all applicable standards for the ethics of experimentation and research integrity.

Reviewer #2: Summary:

The authors take advantage of a randomized clinical trial to conduct a nested cohort study at two locations to look at short-term mortality among patients initiating ART. The primary focus of the study is to estimate short-term mortality risk and identify predictors of increased mortality risk at these two locations. They separately analyze those who are enrolled but do not initiate ART from those that did initiate ART. They also looked at reported cause of death in order to exclude some non-HIV related competing causes (injuries).

General comments:

The analysis is well designed to answer the research question at hand. The authors take care to address ‘boundary conditions’ such as factors that could affect uptake of ART in addition to looking at attrition. The authors address differences in outcomes between the two study sites, and they systematically and thoroughly present their analysis from demographics, to bivariate to multivariable and stratified analyses.

Short-term mortality after ART initiation is fairly well understood through analyses of population cohorts (e.g. Yiannoutsos et al. Estimated mortality of adult HIV-infected patients starting treatment with combination antiretroviral therapy Sex Transm Infect 2012;88:i33-i43 – cited in manuscript). However, this analysis provides more detailed analysis of association between baseline resistance and other clinical characteristics such as BMI and exposure to the clinical trial intervention and provides some disaggregation by study setting which may be of specific interest to Kenya. Given only two sites were included it is hard to generalize to rural vs urban effects.

The primary limitations are limited geographic representativeness and age of the dataset. The analysis is dated – with follow-up apparently ending in 2015.

Specific comments:

Methods:

Line 125 – how was cause-of-death ascertained?

Line 187 – not clear what a ‘known mortality’ is, could be explained/defined further. Known to whom?

Discussion:

Rates of LTFU are about half those reported for East Africa from IeDEA, presumably this is this due to more intensive follow-up employed in the study (monthly visits, community health worker follow-up at home, etc), would be good to make this point as it makes mortality estimates more robust.

6. PLOS authors have the option to publish the peer review history of their article (what does this mean?). If published, this will include your full peer review and any attached files.

Reviewer #1: No

Reviewer #2: No

---

## [Author Response · Author response to Decision Letter 0]

17 Sep 2019

Editor Comments and Responses:

There are just a few comments from reviewer #2, which I would encourage you to address in a revision. In addition to those comments, I would make the following suggestions:

1. As this is a cohort study (nested within RCT) please ensure that reporting conforms with the STROBE guidelines (available at http://www.equator-network.org/reporting-guidelines/strobe/). Please submit checklist with revised submission.

This checklist, with the locations (page numbers) in the manuscript with tracked changes, is included in our submission. Clarifying language has been added throughout the manuscript to ensure checklist items are clearly met. The title has also been changed to include the study design per checklist guidelines.

2. In the abstract, I think it would be more interesting to present the results of the multivariable Cox regression analysis as opposed to the unadjusted analyses.

Thank you for this suggestion. Results have been added to the abstract describing the results of the multivariable regression analyses in addition to the unadjusted analyses. Please see the abstract in the manuscript with tracked changes.

3. I wasn't sure of your definition of PDR as an explanatory variable for the regression analyses. Was this based on OLA (i.e. just the mutations detected by OLA) or based on Illumina NGS (i.e. any drug resistance mutation)? It would be helpful to make this clearer in the Methods section.

Thank you. Clarifying language has been added in lines 179-182:

“PDR was defined as having mutations detected by OLA. To prevent false-positives, low-level mutations <25% of an individual’s HIV quasispecies were confirmed using Illumina sequencing described elsewhere [29]. Mutations detected by OLA but not confirmed via Illumina were defined as wild type.”

4. I couldn't see clear information in the Methods section about how missing data were handled in the regression analysis. It looks from the table footnotes that there may have been only few missing values for the variables included in the analysis, but it would still be helpful to explain clearly how you handled this.

Thank you. Clarifying language has been added to lines 253-254:

“Those with missing data were excluded from the regression analyses in which those variables were included.”

Reviewer #2 - Specific comments:

Methods:

Line 125 – how was cause-of-death ascertained?

This was previously described in the manuscript on lines 197-199 in the document with tracked changes, so no language has been added:

“Dates and causes of illnesses, hospitalizations, and deaths were obtained during follow-up from medical records and/or verbal autopsy via a patient’s relative or other contact when available.”

Line 187 – not clear what a ‘known mortality’ is, could be explained/defined further. Known to whom?

Thank you for this suggestion. Known mortalities refers to deaths reported in the study, as opposed to those who were lost to follow-up whose vital status is unknown. Clarifying language has been added to lines 296-297:

“…and the remaining 28 were lost to follow-up with unknown vital status.”

Discussion:

Rates of LTFU are about half those reported for East Africa from IeDEA, presumably this is this due to more intensive follow-up employed in the study (monthly visits, community health worker follow-up at home, etc), would be good to make this point as it makes mortality estimates more robust.

Thank you for this suggestion. Additional language discussing our study along with the IeDEA cohort was added to the discussion on lines 493-498:

“For example, only 5% of participants were lost to follow-up within 12 months in our study, which is much lower than the 20% within 6 months and 10% within 6-12 months of initiating ART reported in the large cohort in Kenya within the International Epidemiologic Databases to Evaluate AIDS (IeDEA) Collaboration [8]. High retention in our study was likely due to intensive follow-up and contributes to more precise and robust mortality risk estimates in our study.”

---

## [Editor Report · Decision Letter 1]

23 Sep 2019

Predictors of mortality within the first year of initiating antiretroviral therapy in urban and rural Kenya: a prospective cohort study

PONE-D-19-18343R1

Dear Dr. Silverman,

We are pleased to inform you that your manuscript has been judged scientifically suitable for publication and will be formally accepted for publication once it complies with all outstanding technical requirements.

With kind regards,

Richard John Lessells, BSc, MBChB, MRCP, DTM&H, DipHIVMed, PhD

Academic Editor

PLOS ONE
---

## [Editor Report · Acceptance letter]

26 Sep 2019

PONE-D-19-18343R1 

Predictors of mortality within the first year of initiating antiretroviral therapy in urban and rural Kenya: a prospective cohort study 

Dear Dr. Silverman:

I am pleased to inform you that your manuscript has been deemed suitable for publication in PLOS ONE. Congratulations! Your manuscript is now with our production department. 

With kind regards,

on behalf of

Dr. Richard John Lessells 

Academic Editor

PLOS ONE